# ACCELERATING REINFORCEMENT LEARNING THROUGH GPU ATARI EMULATION

## ABSTRACT

We introduce CuLE (CUDA Learning Environment), a CUDA port of the Atari Learning Environment (ALE) which is used for the development of deep reinforcement algorithms. CuLE overcomes many limitations of existing CPU-based emulators and scales naturally to multiple GPUs. It leverages GPU parallelization to run thousands of games simultaneously and it renders frames directly on the GPU, to avoid the bottleneck arising from the limited CPU-GPU communication bandwidth. CuLE generates up to 155M frames per hour on a single GPU, a finding previously achieved only through a cluster of CPUs. Beyond highlighting the differences between CPU and GPU emulators in the context of reinforcement learning, we show how to leverage the high throughput of CuLE by effective batching of the training data, and show accelerated convergence for A2C+V-trace. CuLE is available at [URL revealed upon acceptance].

## 1 INTRODUCTION

Initially triggered by the success of DQN Mnih et al. (2015), research in Deep Reinforcement Learning (DRL) has grown in popularity in the last years Lillicrap et al. (2015); Mnih et al. (2016; 2015), leading to intelligent agents that solve non-trivial tasks in complex environments. But DRL also soon proved to be a challenging computational problem, especially if one wants to achieve peak performance on modern architectures.

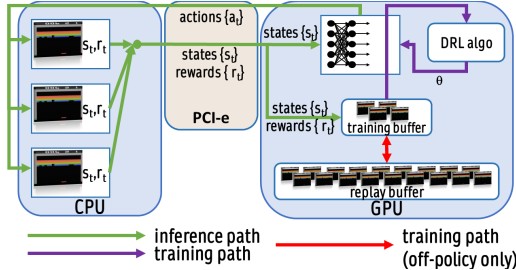

Fig. 1 is a schematic representation of a modern DRL algorithm implementation: CPU environments execute a set of actions $\{a_{t-1}\}$ at time $t-1$, and produce observable states $\{s_t\}$ and rewards $\{r_t\}$. These data are migrated to a Deep Neural Network (DNN) on the GPU to eventu-

Figure 1: In a typical DRL system, environments run on CPUs and DNNs on GPUs. The limited CPU-GPU communication bandwidth and small set of environments prevent full GPU usage.

ally select the next action set, $\{a_t\}$, which is copied back to the CPU. This sequence of operations defines the *inference path*, whose main aim is to generate training data. A training buffer on the GPU stores the states generated on the *inference path*; this is periodically used to update the DNN's weights $\theta$, according to the training rule of the DRL algorithm (*training path*). A computationally efficient DRL system should balance the data generation and training processes, while minimizing the communication overhead along the *inference path* and consuming, along the *training path*, as many data per second as possible Babaeizadeh et al. (2016; 2017). The solution to this problem is however non-trivial and many DRL implementations do not leverage the full computational potential of modern systems Stooke & Abbeel (2018b).

We focus our attention on the *inference path* and move from the traditional CPU implementation of the Atari Learning Environment (ALE), a set of Atari 2600 games that emerged as an excellent DRL benchmark Bellemare et al. (2013); Machado et al. (2017). We show that significant performance bottlenecks primarily stem from CPU environment emulation: the CPU cannot run a large set of

environments simultaneously, the CPU-GPU communication bandwidth is limited, and the GPU is consequently underutilized. To both investigate and mitigate these limitations we introduce CuLE (CUDA Learning Environment), a DRL library containing a CUDA enabled Atari 2600 emulator, that renders frames directly in the GPU memory, avoids off-chip communication and achieves high GPU utilization by processing thousands of environments in parallel—something so far achievable only through large and costly distributed systems. Compared to the traditional CPU-based approach, GPU emulation improves the utilization of the computational resources: CuLE on a single GPU generates more Frames Per Second[1] (FPS) on the *inference path* (between 39K and 125K, depending on the game, see Table 1) compared to its CPU counterpart (between 12.5K and 19.8K). Beyond offering CuLE ([URL revealed upon acceptance]) as a tool for research in the DRL field, our contribution can be summarized as follow:

**(1)** We identify common computational bottlenecks in several DRL implementations that prevent effective utilization of high throughput compute units and effective scaling to distributed systems.

**(2)** We introduce an effective batching strategy for large environment sets, that allows leveraging the high throughput generated by CuLE to quickly reach convergence with A2C+V-trace Espeholt et al. (2018), and show effective scaling on multiple GPUs. This leads to the consumption of 26-68K FPS along the *training path* on a single GPU, and up to 187K FPS using four GPUs, comparable (Table 1) to those achieved by large clusters Stooke & Abbeel (2018a); Espeholt et al. (2018).

**(3)** We analyze advantages and limitations of GPU emulation with CuLE in DRL, including the effect of thread divergence and of the lower (compared to CPU) number of instructions per second per thread, and hope that our insights may be of value for the development of efficient DRL systems.

Table 1: Average training times, raw frames to reach convergence, FPS, and computational resources of existing accelerated DRL schemes, compared to CuLE. Data from Horgan et al. (2018); FPS are taken from the corresponding papers, if available, and measured on the entire Atari suite for CuLE.

| Algorithm | Time | Frames | FPS | Resources | Notes |
|---|---|---|---|---|---|
| Ape-X DQN Horgan et al. (2018) | 5 days | 22,800M | 50K | 376 cores, 1 GPU | — |
| Rainbow Hessel et al. (2017) | 10 days | 200M | — | 1 GPU | — |
| Distributional (C51) Bellemare et al. (2017) | 10 days | 200M | — | 1 GPU | — |
| A3C Mnih et al. (2016) | 4 days | — | 2K | 16 cores | — |
| GA3C Babaeizadeh et al. (2016; 2017) | 1 day | — | 8K | 16 cores, 1 GPU | — |
| Prioritized Dueling Wang et al. (2015) | 9.5 days | 200M | — | 1 GPU | — |
| DQN Mnih et al. (2015) | 9.5 days | 200M | — | 1 GPU | — |
| Gorila DQN Nair et al. (2015) | 4 days | — | — | > 100 cores | — |
| Unreal Jaderberg et al. (2016) | — | 250M | — | 16 cores | — |
| Stooke (A2C / DQN) Stooke & Abbeel (2018b) | hours | 200M | 35K | 40 CPUs, 8 GPUs (DGX-1) | — |
| IMPALA (A2C + V-Trace) Espeholt et al. (2018) | mins/hours | 200M | 250K | 100-200 cores, 1 GPU | — |
| CuLE (*emulation only*) | N/A | N/A | 41K-155K | System I (1 GPU) | 4096 ALEs |
| CuLE (*inference only*, A2C, single batch) | N/A | N/A | 39K-125K | System I (1 GPU) | 4096 ALEs |
| CuLE (*training*, A2C + V-trace, multiple batches) | 1 hour | 200M | 26K-68K | System I (1 GPU) | 1200 ALEs |
| CuLE (*training*, A2C + V-trace, multiple batches)* | mins | 200M | 142-187K | System III (4 GPUs) | 1200×4 ALEs |

*FPS measured on Asterix, Assault, MsPacman, and Pong.

## 2 RELATED WORK

The wall clock convergence time of a DRL algorithm is determined by two main factors: its *sample efficiency*, and the *computational efficiency* of its implementation. Here we analyze the sample and computational efficiency of different DRL algorithms, in connection with their implementation.

Table 2: Systems used for experiments.

| System | Intel CPU | NVIDIA GPU |
|---|---|---|
| I | 12-core Core i7-5930K @3.50GHz | Titan V |
| II | 6-core Core i7-8086K @5GHz | Tesla V100 |
| III | 20-core Core E5-2698 v4 @2.20GHz × 2 | Tesla V100 × 8, NVLink |

---

[1]Raw frames are reported here and in the rest of the paper, unless otherwise specified. These are the frames that are actually emulated, but only 25% of them are rendered and used for training. Training frames are obtained dividing the raw frames by 4—see also Espeholt et al. (2018).

We first divide DRL algorithms into policy gradient and Q-value methods, as in Stooke & Abbeel (2018b). Q-learning optimizes the error on the estimated action values as a proxy for policy optimization, whereas policy gradient methods directly learn the relation between a state, $s_t$, and the optimal action, $a_t$; since at each update they follow, by definition, the gradient with respect to the policy itself, they improve the policy more efficiently. Policy methods are also considered more general, e.g. they can handle continuous actions easily. Also the on- or off-policy nature of an algorithm profoundly affects both its sample and computational efficiency. Off-policy methods allow re-using experiences multiple times, which directly improves the sample efficiency; additionally, old data stored in the GPU memory (replay buffer in Fig. 1) can be used to continuously update the DNN on the GPU, leading to high GPU utilization without saturating the *inference path*. The replay buffer has a positive effect on the stability of learning as well Mnih et al. (2015). On-policy algorithms saturate the *inference path* more easily, as frames have to be generated on-the-fly using the current policy and moved from the CPU emulators to the GPU for processing with the DNN. On-policy updates are generally effective but they are also more prone to fall into local minima because of noise, especially if the number of environment is small — this is the reason why on-policy algorithms largely benefit (in term of stability) from a significant increase of the number of environments.

Among the policy gradient methods, A3C Mnih et al. (2016) is an asynchronous, on-policy, actor-critic algorithm where 16 CPU agents use copies of the same DNN to interact with CPU environments; $N$-step bootstrapping is used to reduce the variance of the critic, $V(s_t; \theta)$: the agents send updates to the DNN after every $N = 5$ actions, and the new set of global $\theta$ weights are broadcast to all agents after each update. A3C can solve an Atari game in approximately 4 days; its hybrid CPU-GPU implementation (GA3C Babaeizadeh et al. (2016; 2017)) stores a single DNN on the GPU, collects CPU data in a system of queues to send them to the GPU in large batches, and achieves convergence in 1 day. Recent (and faster) policy gradient implementations, like PAAC Clemente et al. (2017), PPO Schulman et al. (2017), or A2C OpenAI (2017), store training data directly in a GPU training buffer, as depicted in Fig. 1, which is well representative of the flow of computation of modern DRL algorithms.

Policy gradient algorithms are often on-policy: their efficient update strategy is counterbalanced by the bottlenecks in the *inference path* and competition for the use of the GPU along the *inference* and *training path* at the same time. Acceleration by scaling to a distributed system is possible but inefficient in this case: in IMPALA Espeholt et al. (2018) a cluster with hundreds of CPU cores is needed to accelerate A2C, while training is desynchronized to hide latency. As a consequence, the algorithm becomes off-policy, and V-trace was introduced to deal with off-policy data (see details in the Appendix). Acceleration on a DGX-1 has also been demonstrated for A2C and PPO, using large batch sizes to increase the GPU occupancy, and asynchronous distributed models that hide latency, but require periodic updates to remain synchronized Stooke & Abbeel (2018b) and overall achieves sublinear scaling with the number of GPUs. In practice, the efficiency generally scales sub-linearly on distributed systems, and though Atari games can be solved in a few hours or even minutes, the cost of such systems makes them prohibitive for many researchers.

Q-value methods, like DQN Mnih et al. (2015) and its more recent, improved versions (DoubleDQN van Hasselt et al. (2015), Dueling Networks Wang et al. (2015), Prioritized Replay Schaul et al. (2015), n-step learning Peng & Williams (1996), NoisyNets Fortunato et al. (2017), Rainbow Hessel et al. (2017)) are generally off-policy and can be accelerated more easily, as the *inference* and *training path* can be decoupled, with the GPU continuously fed with training data from the replay buffer in Fig.1. Mapping the two paths on different devices is natural and effective, and indeed DQN methods have been significantly accelerated on distributed systems (see Nair et al. (2015) and Jaderberg et al. (2016) in Table 1). The replay buffer increases the sample efficiency, but the overall convergence time has to pay the cost of using a Q-value method. Furthermore, implementation on large scale systems still incurs a communication overhead and thus generally achieves sub-linear scaling on large distributed systems.

The best compromise between sample efficiency, computational efficiency and effectiveness of the DNN update seems to be achieved by using an on-policy while correcting for stale data, gradient policy methods like A2C+V-trace, which is therefore the algorithm analyzed in detail here, but still suffers from computational bottlenecks in its CPU distributed implementation Espeholt et al. (2018).

**Other Data Generation Engines**   Frameworks to generate training experiences already exist, like the well-known OpenAI Gym, built on top of the C++ Stella emulator Brockman et al. (2016) for Atari games. Other frameworks that target specific problems, such as navigation Mirowski et al. (2016), aim at creating large benchmark for DRL Tassa et al. (2018), or accelerate simulation on the CPU with optimized C++ multi-threaded implementations, while also providing large, GPU-friendly batches for inference and training Tian et al. (2017). To the best of our knowledge, however, none of these attempts directly address the problem of optimizing the *inference path*, i.e., trying to increase the number of environments while avoiding the CPU and bandwidth bottlenecks, by providing a direct implementation of the environments on the GPU.

## 3   CUDA LEARNING ENVIRONMENT (CuLE)

Despite the number and variety of games developed for the Atari 2600, it is relatively simple to emulate the functioning of its hardware much faster than real-time. The Atari console has a 1.19Mhz CPU (to execute the game instructions) and a 128 bytes RAM (containing the game state); the code of each game is stored in a cartridge ROM (typically 2–4kB). The instruction set of each game contains both game and rendering instructions; the first ones are executed by the CPU, while the second ones are sent to the Television Interface Adaptor (TIA), a secondary processor embedded in the Atari hardware that executes the rendering instructions to update a limited set of registers and render the $160 \times 210$ game screen using a 128-colour palette.

In CuLE, we emulate the functioning of many Atari consoles in parallel using the CUDA programming model, where a sequential host program executes parallel programs, known as kernels, on a GPU. In a trivial mapping of the Atari emulator to CUDA, a single thread emulates both the Atari CPU and TIA to execute the ROM

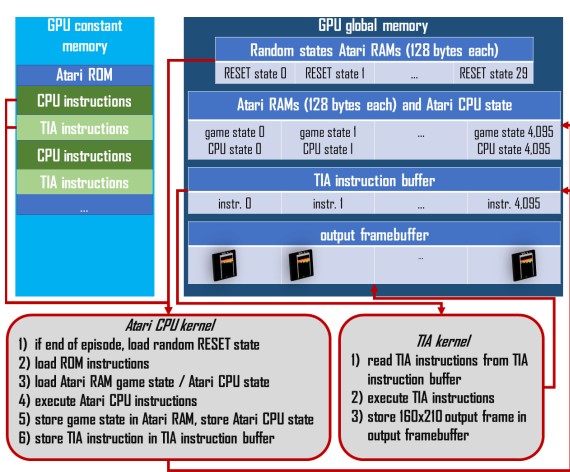

Figure 2: Our CUDA-based Atari emulator uses an *Atari CPU kernel* to emulate the functioning of the Atari CPU and advance the game state, and a second *TIA kernel* to emulate the TIA and render frames directly in GPU memory. For episode resetting we generate and store a cache of random initial states. Massive parallelization on GPU threads allows the parallel emulation of thousands of Atari games.

code, update the Atari CPU and TIA registers as well as the game state in the 128 bytes RAM, and eventually render the pixels in the output frame. However, the contrasting nature of the game code execution and renderings tasks, the first dominated by reading from the RAM/ROM and writing tens of bytes to RAM, while the second writes hundreds of pixels to the framebuffer, poses a serious issue in terms of performance, such as thread divergence and an imbalanced number of registers required by the first and second tasks. To mitigate these issues, CuLE uses two CUDA kernels: the first one first loads data from the GPU global memory, where we store the state of each emulated Atari processor, and the 128 bytes RAM data containing the current state of the game; it also reads ROM instructions from the constant GPU memory, executes them to update the Atari CPU and game states, and stores the updated Atari CPU and game states back into GPU global memory. It is important to notice that this first kernel does not execute the TIA instructions read from the ROM, but copies them into the TIA instruction buffer in GPU global memory, which we implemented to decouple the execution of the Atari CPU and TIA instructions in CuLE. The second CuLE kernel emulates the functioning of the TIA processor: it first reads the instructions stored in the TIA instruction buffer, execute them to update the TIA registers, and renders the $160 \times 210$ output framebuffer in global GPU memory. Despite this implementation requires going through the TIA instruction twice, it has several advantages over the single-kernel trivial implementation. First

of all, the requirements in terms of registers per thread and the chance of having divergent code are different for the Atari CPU and TIA kernels, and the use of different kernels achieves a better GPU usage. A second advantage that we exploit is that not all frames are rendered in ALE: the input of the RL algorithm is the pixelwise maximum between the last two frames in a sequence of four, so we can avoid calling the TIA kernel when rendering of the screen is not needed. A last advantage, not exploited in our implementation yet, is that the TIA kernel may be scheduled one the GPU with more than one thread per game, as rendering of diverse rows on the screen is indeed a parallel operation - we leave this optimization for future developments of CuLE.

To better fit our execution model, our game reset strategy is also different from the one in the existing CPU emulators, where 64 startup frames are executed at the end of each episode. Furthermore, wrapper interfaces for RL, such as ALE, randomly execute an additional number of frames (up to 30) to introduce randomness into the initial state. This results into up to 94 frames to reset a game, which may cause massive divergence between thousands of emulators executing in SIMD fashion on a GPU. To address this issue, we generate and store a cache of random initial states (30 by default) when a set of environments is initialized in CuLE. At the end of an episode, each emulator randomly selects one of the cached states as a seed and copies it into the terminal emulator state.

Some of the choices made for the implementation of CuLE are informed by ease of debugging, like associating one state update kernel to one environment, or need for flexibility, like emulating the Atari console instead of directly writing CUDA code for each Atari game. A 1-to-1 mapping between threads and emulators is not the most computationally efficient way to run Atari games on a GPU, but it makes the implementation relatively straightforward and has the additional advantage that the same emulator code can be executed on the CPU for debugging and benchmarking (in the following, we will refer to this implementation as CuLE$_{CPU}$). Despite of this, the computational advantage provided by CuLE over traditional CPU emulation remains significant, as shows in the next Section.

## 4 EXPERIMENTS

**Atari emulation**  We measure the FPS under different conditions: we get an upper bound on the maximum achievable FPS in the *emulation only* case, when we emulate the environments and use a random policy to select actions. In the *inference only* case, we measure the FPS along the *inference path*: a policy DNN selects the actions, CPU-GPU data transfer occur for CPU emulators, while both emulation and DNN inference run on the GPU when CuLE is used. This is the maximum throughput achievable by off-policy algorithms, when data generation and consumption are decoupled and run on different devices. In the *training* case, the entire DRL system is at work: emulation, inference, and training may all run on the same GPU. This is representative of the case of on-policy algorithms, but the FPS are also affected by the computational cost of the specific DRL update algorithm; in our experiments we use a vanilla A2C OpenAI (2017), with N-step bootstrapping, and $N = 5$ as the baseline (for details of A2C and off-policy correction with V-trace, see the Appendix).

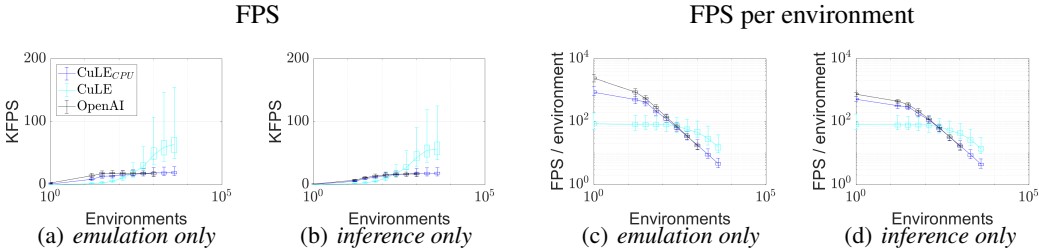

Figure 3: FPS and FPS / environment on System I in Table 2, for OpenAI Gym OpenAI (2017), CuLE$_{CPU}$, and CuLE, as a function of the number of environments, under different load conditions: *emulation only*, and *inference only*. The boxplots indicate the minimum, $25^{th}$, $50^{th}$, $75^{th}$ percentiles and maximum FPS, for the entire set of 57 Atari games.

Table 3: Training FPS, DNN's Update Per Second (UPS), time to reach a given score, and corresponding number of training frames for four Atari games, A2C+V-trace, and different configurations of the emulation engines, measured on System I in Table 2 (System III for the multi-GPU case). The best metric in each row is in bold.

| Engine | OpenAI Gym | | | | CuLE, 1 GPU | | | CuLE, 4 GPUs | Game |
|---|---|---|---|---|---|---|---|---|---|
| Envs | 120 | 120 | 120 | 1200 | 1200 | 1200 | 1200 | 1200×4 | |
| Batches | 1 | 5 | 20 | 20 | 1 | 5 | 20 | 20×4 | |
| N-steps | 5 | 5 | 20 | 20 | 5 | 5 | 20 | 20 | |
| SPU | 5 | 1 | 1 | 1 | 5 | 1 | 1 | 1 | |
| Training KFPS | 4.2 | 3.4 | 3.0 | 4.9 | 10.6 | 11.5 | 11.0 | **42.7** | Assault |
| UPS | 7.0 | **28.3** | 24.7 | 4.1 | 1.8 | 9.6 | 9.1 | 8.9 | |
| Time [mins] | 20.2 | — | 42.6 | 44.2 | 18.8 | 9.4 | 9.9 | **7.9** | |
| Training Mframes (for average score: 800) | **5.0** | — | 7.5 | 13.0 | 12.0 | 6.5 | 6.5 | 18.0 | |
| Training KFPS | 4.3 | 3.3 | 3.0 | 4.9 | 11.9 | 12.5 | 12.1 | **46.6** | Asterix |
| UPS | 7.1 | **27.9** | 24.8 | 4.1 | 2.0 | 10.4 | 10.0 | 9.7 | |
| Time [mins] | 8.1 | 35.2 | 14.4 | 27.1 | — | 14.0 | 3.4 | **2.5** | |
| Training Mframes (for average score: 1,000) | **2.0** | 7.0 | 2.5 | 8.0 | — | 10.5 | 2.5 | 7.0 | |
| Training KFPS | 4.0 | 3.3 | 2.8 | 4.8 | 9.0 | 9.6 | 9.2 | **35.5** | MsPacman |
| UPS | 6.7 | **27.1** | 23.7 | 4.0 | 1.5 | 8.0 | 7.7 | 7.4 | |
| Time [mins] | 16.6 | 20.5 | 14.7 | 12.4 | — | 6.9 | 11.8 | **2.4** | |
| Training Mframes (for average score: 1,500) | 4.0 | 4.0 | **2.5** | 3.5 | — | 4.0 | 6.5 | 3.0 | |
| Training KFPS | 4.3 | 3.4 | 3.0 | 4.8 | 10.5 | 11.2 | 10.6 | **41.7K** | Pong |
| UPS | 7.2 | **28.1** | 24.9 | 4.0 | 1.8 | 9.3 | 8.9 | 8.7 | |
| Time [mins] | 21.2 | 12.2 | 8.4 | 8.7 | — | 5.9 | 3.1 | **2.4** | |
| Training Mframes (for average score: 18) | 5.5 | 2.5 | **1.5** | 2.5 | — | 4.0 | 2.0 | 6.0 | |

Figs. 3(a)-3(b) show the FPS generated by OpenAI Gym, $\text{CuLE}_{\text{CPU}}$, and CuLE, on the entire set of Atari games, as a function of the number of environments. In the *emulation only* case, CPU emulation is more efficient for a number of environments up to 128, when the GPU computational power is not leveraged because of the low occupancy. For a larger number of environments, CuLE significantly overcomes OpenAI Gym, for which FPS are mostly stable for 64 environments or more, indicating that the CPU is saturated: the ratio between the median FPS generated by CuLE with 4096 environment (64K) and the peak FPS for OpenAI Gym (18K) is $3.56\times$. In the *inference only* case there are two additional overheads: CPU-GPU communication (to transfer observations), and DNN inference on the GPU. Consequently, CPU emulators achieve a lower FPS in *inference only* when compared to *emulation only*; the effects of the overheads is more evident for a small number of environments, while the FPS slightly increase with the number of environments without reaching the *emulation only* FPS. CuLE's FPS are also lower for *inference only*, because of the latency introduced by DNN inference, but the FPS grow with the number of environments, suggesting that the computational capability of the GPU is still far from being saturated.

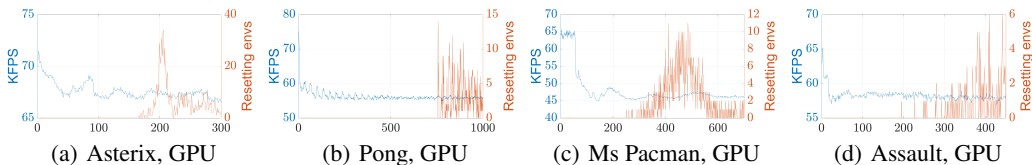

|  |  |  |  |
|---|---|---|---|
| (a) Asterix, GPU | (b) Pong, GPU | (c) Ms Pacman, GPU | (d) Assault, GPU |

Figure 4: FPS as a function of the environment step, measured on System I in Table 2 for *emulation only* on four Atari games, 512 environments, for CuLE; each panel also shows the number of resetting environments. FPS is higher at the beginning, when all environments are in similar states and thread divergence within warps is minimized; after some steps, correlation is lost, FPS decreases and stabilizes. Minor oscillations in FPS are possibly associated to more or less computational demanding phases in the emulation of the environments (e.g., when a goal is scored in Pong).

**Factors affecting the FPS** Figs. 3(a)-3(b) shows that the throughput varies dramatically across games: 4096 $\text{CuLE}_{\text{CPU}}$ environments run at 27K FPS on Riverraid, but only 14K FPS for Boxing: a

$1.93\times$ difference, explained by the different complexity of the ROM code of each game. The ratio between the maximum and minimum FPS is amplified in the case of GPU emulation: Riverraid runs in *emulation only* at 155K FPS when emulated by CuLE and 4096 environments, while UpNDown runs at 41K FPS —a $3.78\times$ ratio.

To better highlight the impact of thread divergence on throughput, we measure the FPS for CuLE, *emulation only*, 512 environments, and four games (Fig. 4). All the environments share the same initial state, but random action selection leads them to diverge after some steps. Each environment resets at the end of an episode. The FPS is maximum at the very beginning, when all the environments are in similar states and the chance to execute the same instruction in all the threads is high. When they move towards different states, code divergence negatively impacts the FPS, until it reaches an asymptotic value. This effect is present in all games and particularly evident for MsPacman in Fig. 4; it is not present in CPU emulation (see Appendix). Although divergence can reduce FPS by $30\%$ in the worst case, this has to be compared with case of complete divergence within each thread and for each instruction, which would yield $1/32 \simeq 3\%$ of the peak performances. Minor oscillations of the FPS are also visible especially for games with a repetitive pattern (e.g. Pong), where different environments can be more or less correlated with a typical oscillation frequency.

**Performances during training**  Fig. 5 compares the FPS generated by different emulation engines on a specific game (Assault)[2], for different load conditions, including the *training* case, and number of environments. As expected, when the entire *training path* is at work, the FPS decreases even further. However, for CPU emulators, the difference between FPS in the *inference only* and *training* cases decreases when the number of environments increases, as the system is bounded by the CPU computational capability and CPU-GPU communication bandwidth. In the case of the CPU scaling to multiple GPUs would be ineffective for on-policy algorithms, such GA3C Babaeizadeh et al. (2016; 2017), or sub-optimal, in the case of distributed systems Espeholt et al. (2018); Stooke & Abbeel (2018b). On the other hand, the difference between *inference only* and *training* FPS increases with the number of environments for CuLE, because of the additional training overhead on the GPU. The potential speed-up provided by CuLE for vanilla A2C and Assault in Fig. 5 is $2.53\times$ for 1,024 environments, but the system is bounded by the GPU computational power; as a consequence, better batching strategies that reduce the training computational overhead as well as scaling to multiple GPUs are effective to further increase the speed-up ratio, as demonstrated later in this Section.

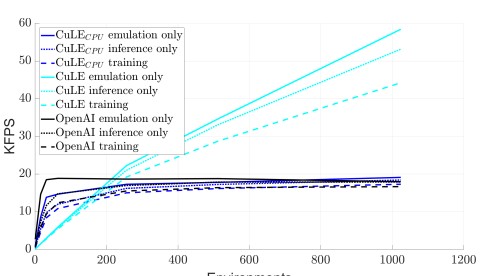

Figure 5: FPS generated by different emulation engines on System I in Table 2 for Assault, as a function of the number of environments, and different load conditions for A2C with N-step bootstrapping, $N = 5$).

When data generation and training can be decoupled, like for off-policy algorithms, training can be easily moved to a different GPU and the *inference path* can be used at maximum speed. The potential speed-up provided by CuLE for off-policy algorithms is then given by the ratio between the *inference only* median FPS for CuLE (56K) and CuLE$_{CPU}$ (18K), which is $3.11\times$ for 4,096 environments. Furthermore, since the FPS remains flat for CPU emulation, the advantage of CuLE amplifies (for both on- and off-policy methods) with the number of environments.

**Frames per second per environment**  Fig. 3(c)-3(d) show the FPS / environment for different emulation engines on System I, as a function of the number of environments. For 128 environments or fewer, CPU emulators generate frames at a higher rate (compared to CuLE), because CPUs are optimized for low latency, and execute a high number of instructions per second per thread. However, the FPS / environment decrease with the number of environments, that have to share the same CPU cores. Instead, the GPU architecture maximizes the throughput and has a lower number of instructions per second per thread. As a consequence, the FPS / environment is smaller (compared to CPU emulation) for a small number of environments, but they are almost constant up to 512 environments, and starts decreasing only after this point. In practice, CuLE environments provide an

---

[2]Other games for which we observe a similar behavior are reported in the Appendix, for sake of space.

efficient means of training with a diverse set of data and collect large statistics about the rewards experienced by numerous agents, and consequently lowering the variance of the value estimate. On the other hand, samples are collected less efficiently in the temporal domain, which may worsen the bias on the estimate of the value function by preventing the use of large N in N-step bootstrapping. The last paragraph of this Section shows how to leverage the high throughput generated by CuLE, considering these peculiarities.

**Memory limitations**    Emulating a massively large number of environments can be problematic considering the relatively small amount of GPU DRAM. Our PyTorch Paszke et al. (2017) implementation of A2C requires each environment to store 4 84x84 frames, plus some additional variables for the emulator state. For 16K environments this translates into 1GB of memory, but the primary issue is the combined memory pressure to store the DNN with 4M parameters and the meta-data during training, including the past states: training with 16K environments easily exhausts the DRAM on a single GPU (while training on multiple GPUs increases the amount of available RAM). Since we did not implement any data compression scheme as in Horgan et al. (2018), we constrain our training configuration to fewer than 5K environments, but peak performance in terms of FPS would be achieved for a higher number of environments - this is left as a possible future improvement.

**A2C**    We analyze in detail the case of A2C with CuLE on a single GPU. As a baseline, we consider vanilla A2C, using 120 OpenAI Gym CPU environments that send training data to the GPU to update the DNN (Fig. 6(a)) every $N = 5$ steps. This configuration takes, on average, 21.2 minutes (and 5.5M training frames) to reach a score of 18 for Pong and 16.6 minutes (4.0M training frames) for a score of 1,500 on Ms-Pacman (Fig. 7, red line; first column of Table 3). CuLE with 1,200 environments generates approximately $2.5\times$ more FPS compared to OpenAI Gym, but this alone is not sufficient to improve the convergence speed (blue line, Fig. 7). CuLE generates larger batches but, because FPS / environment is lower when compared to CPU emulation, fewer Updates Per Second (UPS) are performed for training the DNN (Table 3), which is detrimental for learning.

**A2C+V-trace and batching strategy**    To better leverage CuLE, and similar in spirit to the approach in IMPALA Espeholt et al. (2018), we employ a different batching strategy, illustrated in Fig. 6(b): environment steps occur in parallel on the GPU, but training data are read in batches to update the DNN every Steps Per Update (SPU) steps. This batching strategy significantly increases the DNN's UPS at the cost of a slight decrease in FPS (second columns of OpenAI Gym and CuLE in Table 3), due to the fact that the GPU has to dedicate more time to training. Furthermore, as only the most recent data in a batch are generated with the current policy, we use V-trace Espeholt et al. (2018) for off-policy correction. The net result is an increase of the overall training time when 120

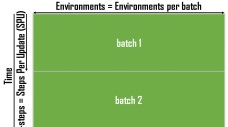 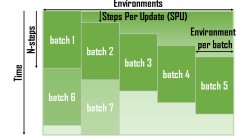

(a) Single batch strategy    (b) Multi batch strategy

Figure 6: Different batching strategies are defined by the number of batches, N-Steps and Steps Per Update (SPU) parameters. The on-policy single batch case (panel a) is a special case of the more general, off-policy multi batch approach (panel b). The batching strategy affects both the computational and convergence aspects of the DRL algorithm, as shown in Fig. 7 and in Table 3.

OpenAI Gym CPU environments are used, as this configuration pays for the increased training and communication overhead, while the smaller batch size increases the variance in the estimate of the value function and leads to noisy DNN updates (second column in Table 3, orange lines in Fig. 7). Since CuLE does not suffer from the same computational bottlenecks, and at the same time benefits from the variance reduction associated with the large number (1,200) of environments, using the same batching strategy with CuLE reduces the time to reach a score of 18 for Pong and 1,500 for Pacman respectively to 5.9 and 6.9 minutes. The number of frames required to reach the same score is sometimes higher for CuLE (Table 3), which can lead to less sample efficient implementation when compared to the baseline, but the higher FPS largely compensates for this. Extending the batch size in the temporal dimension (N-steps bootstrapping, $N = 20$) increases the GPU computational load and reduces both the FPS and UPS, but it also reduces the bias in the estimate of the value function, making each DNN update more effective, and leads to an overall decrease of the

wall clock training time, the fastest convergence being achieved by CuLE with 1,200 environments. Using OpenAI Gym with the same configuration results in a longer training time, because of the lower FPS generated by CPU emulation.

**Scaling to multiple GPUs** The black line in Fig. 7 represents the case of A2C+V-Trace with CuLE and 4 GPUs, each emulating 1,200 environments. The FPS grow almost linearly with the number of GPUs under this *training* load: CuLE running on 4 GPUs allows generating and consuming a comparable number of FPS with respect to IMPALA on a large CPU cluster (Table 1)). This leads to a dramatic reduction of the convergence time, as documented in Table 3, with Pong almost solved in less than 3 minutes.

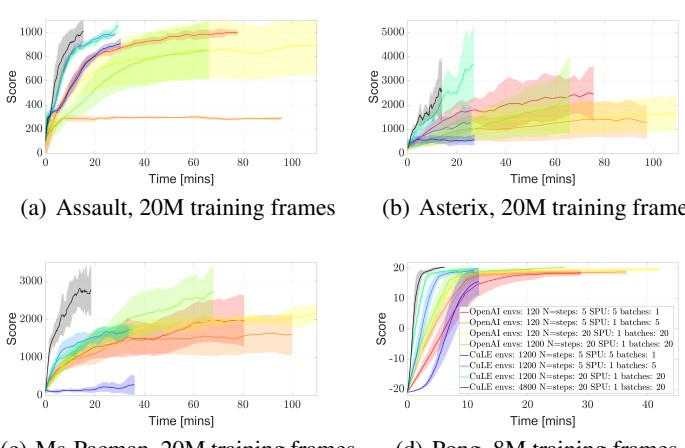

(a) Assault, 20M training frames   (b) Asterix, 20M training frames

(c) Ms-Pacman, 20M training frames   (d) Pong, 8M training frames

Figure 7: Average testing score and standard deviation on four Atari games as a function of the training time, for A2C+V-trace, System III in Table 2, and different batching strategies (see also Table 3). Training frames are double for the multi-GPU case (black line). Training performed on CuLE or OpenAI Gym; testing performed on OpenAI Gym environments (see the last paragraph of Section 4).

Beyond A2C+V-trace, we report in Table 4 the time to generate 200M frames for PPO and Rainbow DQN by running CuLE on 1 to 8 GPUs on a single system. We use the PyTorch multiprocessing facilities to launch 1 process for each GPU and update gradients in a distributed manner using the NVIDIA NCCL multi-GPU communication backend. This Table shows efficient scaling of the training FPS with the number of GPUs. As in the case of A2C+V-trace, an efficient batching strategy is then required to fully leverage the high throughput generated by CuLE and can significantly improve these figures.

**Generalization for different systems** Table 5 reports the FPS for the implementations of vanilla DQN, A2C, and PPO, on System I and II in Table 2. The speed-up in terms of FPS provided by CuLE is consistent across different systems, different algorithms, and larger in percentage for a large number of environments. Different DRL algorithms achieve different FPS depending on the complexity and frequency of the training step on the GPU.

Table 4: Hours to complete 50M training frames (200M raw frames) on System III in Table 2, by GPU count, for two different DRL algorithms, 2,048 environments per GPU.

| Algo | 1 GPU | 2 GPUs | 4 GPUs | 8 GPUs |
|---|---|---|---|---|
| PPO | 9.38 | 4.45 | 2.12 | 1.1 |
| Rainbow DQN | 10.5 | 5.1 | 3.2 | 2.1 |

The same Table also reports the minimum and maximum GPU utilization measured while running each DRL algorithm. In its vanilla implementation DQN is characterized by long GPU idle times occurring during CPU emulation, leading to a low GPU utilization. The GPU utilization increases when emulation is moved to the GPU, while GPU peak utilization is reached during the DNN training step. GPU underutilization can be observed also for policy-gradient algorithms (A2C, PPO) and CPU emulation; peak utilization is higher for PPO, because of its larger computational complexity. Nearly full GPU utilization is achieved only by CuLE for a large number of environments.

**Correctness of the implementation**   To guarantee that CuLE is compliant with OpenAI Gym, we seed OpenAI Gym and CuLE environments with the same game state and successfully verify that the sequence of game states and frames returned by each environment are equivalent. A more

Table 5: Average FPS and min/max GPU utilization during training for Pong with different algorithms and using different emulation engines on different systems (see Table 2); CuLE consistently leads to higher FPS and GPU utilization.

| Algorithm | Emulation engine | FPS [GPU utilization %] | | | |
|---|---|---|---|---|---|
| | | System I [256 envs] | System I [1024 envs] | System II [256 envs] | System II [1024 envs] |
| DQN | OpenAI | 6.4K [15-42%] | 8.4K [0-69%] | 10.8K [26-32%] | 21.2K [28-75%] |
| | CuLE$_{CPU}$ | 7.2K [16-43%] | 8.6K [0-72%] | 6.8K [17-25%] | 20.8K [8-21%] |
| | CuLE | 14.4K [16-99%] | 25.6K [17-99%] | 11.2K [48-62%] | 33.2K [57-77%] |
| A2C | OpenAI | 12.8K [2-15%] | 15.2K [0-43%] | 24.4K [5-23%] | 30.4K [3-45%] |
| | CuLE$_{CPU}$ | 10.4K [2-15%] | 14.2K [0-43%] | 12.8K [1-18%] | 25.6K [3-47%] |
| | CuLE | 19.6K [97-98%] | 51K [98-100%] | 23.2K [97-98%] | 48.0K [98-99%] |
| PPO | OpenAI | 12K [3-99%] | 10.6K [0-96%] | 16.0K [4-33%] | 19.2K [4-62%] |
| | CuLE$_{CPU}$ | 10K [2-99%] | 10.2K [0-96%] | 9.2K [2-28%] | 18.4K [3-61%] |
| | CuLE | 14K [95-99%] | 36K [95-100%] | 14.4K [43-98%] | 28.0K [45-99%] |

subtle question to answer is if the CuLE reset procedure, based on a set of random initial states, can reduce the variability in the environments and consequently be detrimental for the learning and generalization capability of the trained agents. Our code ([URL revealed upon acceptance]) allows the user to test the training agent on CuLE, CuLE$_{CPU}$, and OpenAI Gym environments: in our tests the results are comparable for the three backends. Results reported in Fig. 7 and in the Appendix further suggest that CuLE trained agents do not suffer any performance loss when tested on OpenAI Gym environments.

## 5 DISCUSSION AND CONCLUSION

The common allocation of the tasks in a DRL system dictates that the environment should run on CPUs, whereas GPUs should be dedicated to DNN operations. With most of the existing frameworks Brockman et al. (2016); Mirowski et al. (2016); Tassa et al. (2018); Tian et al. (2017) following this paradigm, the limited CPU-GPU communication bandwidth and CPU capability to emulate a large number of environments represent two limiting factors to effectively accelerate DRL algorithms, even when mapped to expensive distributed systems. By rendering frames directly on the GPU, CuLE overcomes these limitations and generate as many FPS as those generated by large, expensive CPU systems. CuLE promises to be an effective tool to develop and test DRL algorithms by significantly reducing the experiment turnaround time.

As already shown by others in the case of DRL on distributed system, our experiments show that proper batching coupled with a slight off-policy gradient policy algorithm can significantly accelerate the wall clock convergence time; CuLE has the additional advantage of allowing effective scaling of this implementation to a system with multiple GPUs. CuLE effectively allows increasing the number of parallel environments but, because of the low number of instructions per second per thread on the GPU, training data can be narrow in the time direction. This can be problematic for problems with sparse temporal rewards, but rather than considering this as a pure limitation of CuLE, we believe that this peculiarity opens the door to new interesting research questions, like active sampling of important states Hessel et al. (2017); Wang et al. (2015) that can then be effectively analyzed on a large number of parallel environments with CuLE. CuLE also hits a new obstacle, which is the limited amount of DRAM available on the GPU; studying new compression schemes, like the one proposed in Hessel et al. (2017), as well as training methods with smaller memory footprints may help extend the utility of CuLE to even larger environment counts, and design better GPU-based simulator for RL in the future. Since these are only two of the possible research directions for which CuLE is an effective investigation instrument, CuLE comes with a python interface that allows easy experimentation and is freely available to any researcher at [URL revealed upon acceptance].

A last note has to be done on CuLE's implementation, that is informed by ease of debugging, need for flexibility, and compatibility with standard DRL benchmarks. These choices put a limit on the achievable speed up factor (for instance by using emulation of the Atari 2600 console instead of direct CUDA implementations of Atari games), but the analysis and insights provided in our paper furnish indications for the design of efficient simulators for DRL.

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

# A APPENDIX

## A.1 REINFORCEMENT LEARNING, A2C AND V-TRACE

**Reinforcement learning** In RL, an agent observes a state $s_t$ at time $t$ and follows a policy $\pi = \pi(s_t)$ to select an action $a_t$; the agent also receives a scalar reward $r_t$ from the environment. The goal of RL is to optimize $\pi$ such that the sum of the expected rewards is maximized.

In model-free policy gradient methods $\pi(a_t|s_t; \theta)$ is the output of a policy DNN with weights $\theta$, and represents the probability of selecting action $a_t$ in the state $s_t$. Updates to the DNN are generally aligned in the direction of the gradient of $E[R_t]$, where $R_t = \sum_{i=0}^{\infty} \gamma^i r_{t+i}$ is the discounted reward from time $t$, with discount factor $\gamma \in (0, 1]$ (see also REINFORCE Williams (1992)) The vanilla implementation updates $\theta$ along $\nabla_\theta \log \pi(a_t|s_t; \theta) R_t$, which is an unbiased estimator of $\nabla_\theta E[R_t]$. The training procedure can be improved by reducing the variance of the estimator by subtracting a learned *baseline* $b_t(s_t)$ and using the gradient $\nabla_\theta \log \pi(a_t|s_t; \theta)[R_t - b_t(s_t)]$. One common baseline is the value function $V^\pi(s_t) = E[R_t|s_t]$, which is the expected return for the policy $\pi$ starting from $s_t$. The policy $\pi$ and the baseline $b_t$ can be viewed as *actor* and *critic* in an actor-critic architecture Sutton & Barto (1998).

**A2C** A2C OpenAI (2017) is the synchronous version of A3C Mnih et al. (2016), a successful actor-critic algorithm, where a single DNN outputs a softmax layer for the policy $\pi(a_t|s_t; \theta)$, and a linear layer for $V(s_t; \theta)$. In A2C, multiple agents perform simultaneous steps on a set of parallel environments, while the DNN is updated every $t_{max}$ actions using the experiences collected by all the agents in the last $t_{max}$ steps. This means that the variance of the critic $V(s_t; \theta)$ is reduced (at the price of an increase in the bias) by $N$-step bootstrapping, with $N = t_{max}$. The cost function for the policy is then:

$$\log \pi(a_t|s_t; \theta) \left[ \tilde{R}_t - V(s_t; \theta_t) \right] + \beta H \left[ \pi(s_t; \theta) \right], \tag{1}$$

where $\theta_t$ are the DNN weights $\theta$ at time $t$, $\tilde{R}_t = \sum_{i=0}^{k-1} \gamma^i r_{t+i} + \gamma^k V(s_{t+k}; \theta_t)$ is the bootstrapped discounted reward from $t$ to $t + k$ and $k$ is upper-bounded by $t_{max}$, and $H[\pi(s_t; \theta)]$ is an entropy term that favors exploration, weighted by the hyper-parameter $\beta$. The cost function for the estimated value function is:

$$\left[ \tilde{R}_t - V(s_t; \theta) \right]^2, \tag{2}$$

which uses, again, the bootstrapped estimate $\tilde{R}_t$. Gradients $\nabla \theta$ are collected from both of the cost functions; standard optimizers, such as Adam or RMSProp, can be used for optimization.

**V-trace** In the case where there is a large number of environments, such as in CuLE or IM-PALA Espeholt et al. (2018), the synchronous nature of A2C become detrimental for the learning speed, as one should wait for all the environments to complete $t_{max}$ steps before computing a single DNN update. Faster convergence is achieved (both in our paper and in Espeholt et al. (2018)) by desynchronizing data generation and DNN updates, which in practice means sampling a subset of experiences generated by the agents, and updating the policy using an approximate gradient, which makes the algorithm slightly off-policy.

To correct for the off-policy nature of the data, that may lead to inefficiency or, even worse, instabilities, in the training process, V-trace is introduced in Espeholt et al. (2018). In summary, the aim of off-policy correction is to give less weight to experiences that have been generated with policy $\mu$, called the *behaviour policy*, when it differs from the *target policy*, $\pi$; for a more principled explanation we remand the curios reader to Espeholt et al. (2018).

For a set of experiences collected from time $t = t_0$ to time $t = t_0 + N$ following some policy $\mu$, the $N$-steps V-trace target for $V(s_{t_0}; \theta)$ is defined as:

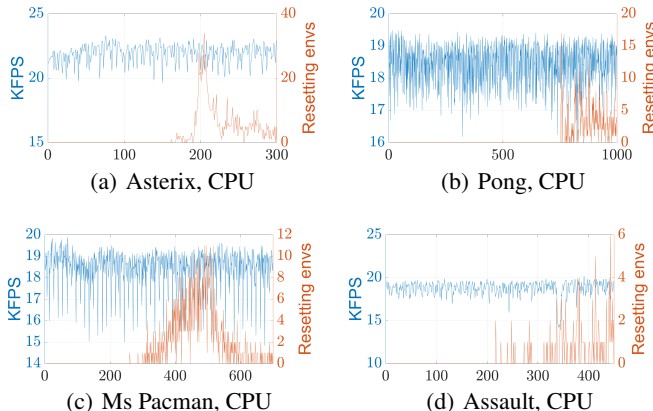

Figure 8: FPS as a function of the environment step, measured on System I in Table 2 for *emulation only* on four Atari games, 512 environments, for CuLE$_{\text{CPU}}$; each panel also shows the number of resetting environments. A peak in the FPS at the beginning of the emulation period, as in the case of GPU emulation in Fig. 4, is not visible in this case.

$$v_{t_0} \;=\; V(s_{t_0};\theta) + \sum_{t=t_0}^{t_0+N-1} \gamma^{t-t_0}\Big(\prod_{i=t_0}^{t-1} c_i\Big)\delta_t V, \tag{3}$$

$$\delta_t V \;=\; \rho_t\big(r_t + \gamma V(s_{t+1};\theta) - V(s_t;\theta)\big) \tag{4}$$

$$\rho_t \;=\; \min\big(\bar{\rho}, \frac{\pi(a_t|s_t)}{\mu(a_t|s_t)}\big) \tag{5}$$

$$c_i \;=\; \min\big(\bar{c}, \frac{\pi(a_i|s_i)}{\mu(a_i|s_i)}\big); \tag{6}$$

$\rho_t$ and $c_i$ are truncated importance sampling (IS) weights, and $\prod_{i=t0}^{t-1} c_i = 1$ for $s = t$, and $\bar{\rho} \geq \bar{c}$. Notice that, when we adopt the proposed multi-batching strategy, there are multiple behaviour policies $\mu$ that have been followed to generate the training data — e.g., N different policies are used when SPU=1 in Fig. 6(b). Eqs. 5-6 do not need to be changed in this case, but we have to store all the $\mu(a_i|s_i)$ in the training buffer to compute the, V-trace corrected, DNN update. In our implementation, we compute the V-trace update recursively as:

$$v_t = V(s_t;\theta) + \delta_t V + \gamma c_s\big(v_{t+1} - V(s_{t+1};\theta)\big). \tag{7}$$

At training time $t$, we update $\theta$ with respect to the value output, $v_s$, given by:

$$\big(v_t - V(s_t;\theta)\big)\nabla_\theta V(s_t;\theta), \tag{8}$$

whereas the policy gradient is given by:

$$\rho_t \nabla_\omega \log \pi_\omega(a_s|s_t)\big(r_t + \gamma v_{t+1} - V(s_t;\theta)\big). \tag{9}$$

An entropy regularization term that favors exploration and prevents premature convergence (as in Eq. 1) is also added.

### A.2    THREAD DIVERGENCE IS NOT PRESENT IN THE CASE OF CPU EMULATION

We show here that thread divergence, that affects GPU-based emulation (see Fig. 4), does not affect CPU-based emulation. Fig. 8 shows the FPS on four Atari games where all the environments share the same initial state. In constrast with GPU emulation, the CPU FPS do not peak at the beginning of the emulation period, where many environments are correlated.

### A.3 PERFORMANCE DURING TRAINING - OTHER GAMES

For sake of space, we only report (Fig. 9) the FPS measured on system I in Table 2 for three additional games, as a function of different load conditions and number of environments.

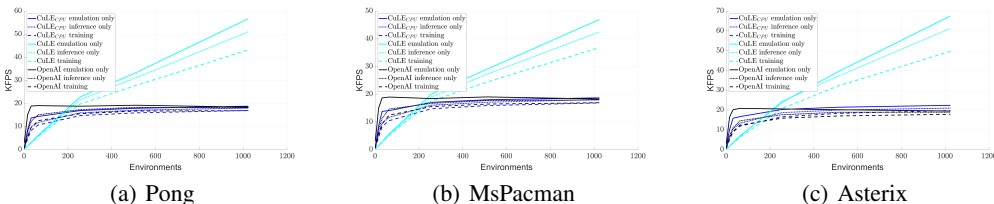

|        (a) Pong        |      (b) MsPacman      |       (c) Asterix       |

Figure 9: FPS generated by different emulation engines on System I in Table 2 for different Atari games, as a function of the number of environments, and different load conditions (the main A2C OpenAI (2017) loop is run here, with N-step bootstrapping, $N = 5$.

### A.4 CORRECTNESS OF THE IMPLEMENTATION

To demonstrate the correctness of our implementation, and thus that policies learned with CuLE generalize to the same game emulated by OpenAI Gym, we report in Fig. 10 the average scores achieved in testing, while training an agent with with A2C+V-trace and CuLE. The testing scores measured on CuLE$_{CPU}$ and OpenAI Gym environments do not show any relevant statistical difference, even for the case of Ms-Pacman, where the variability of the scores is higher because of the nature of the game.

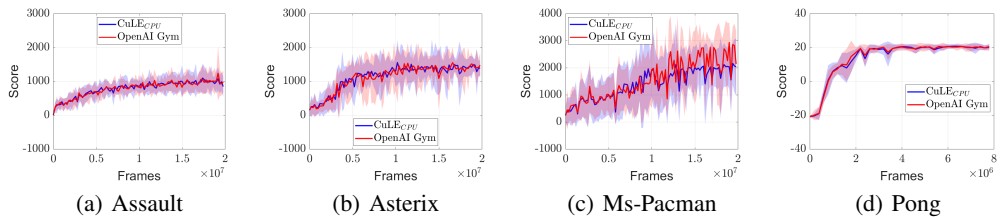

|   (a) Assault   |   (b) Asterix   |   (c) Ms-Pacman   |   (d) Pong   |

Figure 10: Average testing scores measured on 10 CuLE$_{CPU}$ and OpenAI Gym environments, while training with A2C+V-trace and CuLE, as a function of the training frames; 250 environments are used for Ms-Pacman, given its higher variability. The shaded area represents 2 standard deviations.

