# OpenReview forum: "Accelerating Reinforcement Learning Through GPU Atari Emulation"
_ICLR.cc/2020/Conference — Reject_

### Official Review · AnonReviewer3 · 2019-10-22
**Official Blind Review #3**

**Rating:** 8

**Review:**

This paper introduces a CUDA port of the Atari Learning Environment. The paper goes into detail examining the benefits that come from a GPU-only implementation, including much better per-GPU utilization as well as no need to run a distributed system of CPUs. They show this hardware scaling can be taken advantage of across a variety of state of the art reinforcement learning algorithms and indeed create new batching strategies to utilize their framework and the GPU better.

The paper is well written and goes into some detail describing the implementation of CuLE as well as various design decisions taken, as with splitting the emulation process across several kernels. Finally, the paper is very explicit about a number of optimizations that are not being exploited by the new framework and serve as markers for future work.

A question that arises and which is not addressed in the experiments is how the authors verified their port is faithful to the original version; there is no mention of correctness in the paper.

**Experience Assessment:**

I have read many papers in this area.

**Review Assessment: Checking Correctness Of Derivations And Theory:**

I assessed the sensibility of the derivations and theory.

**Review Assessment: Checking Correctness Of Experiments:**

I assessed the sensibility of the experiments.

**Review Assessment: Thoroughness In Paper Reading:**

I read the paper at least twice and used my best judgement in assessing the paper.

---

> ### Author Response · Authors · 2019-11-08
> **Answer to reviewer 3**
>
> The point raised by R3 about the correctness of our implementation (and its faithful adherence to the original implementation of Atari) is indeed an important one and deserves attention. We thank the reviewer for having raised this point. We modified the caption of Fig. 7 and appended a paragraph to Section 4 to describe how we checked the correctness of our implementation: “To guarantee that CuLE is compliant with OpenAI Gym, we seed OpenAI Gym and CuLE environments with the same game state and successfully verified that the sequence of game states and frames returned by each environment are equivalent. A more subtle question to answer is if the CuLE reset procedure, based on a set of random initial states, can reduce the variability in the environments and consequently be detrimental for the learning and generalization capability of the trained agents. Our code released in [CULE URL] allows the user to test the training agent on CuLE, CuLE-CPU, and OpenAI Gym environments: in our tests the results are comparable for the three backends. Results reported in Fig. 7} further suggest that CuLE trained agents do not suffer any performance loss when tested on OpenAI Gym environments.”

---

> > ### Author Response · Authors · 2019-11-14
> > **Additional answer to reviewer 3**
> >
> > To better support our claim about the correctness of our implementation, and provide a more complete set of information to the reviewer and the reader, we further added a last paragraph in the Appendix (and Fig. 10)  to show that evaluation of an agent trained on CuLE data, performed on CuLE-CPU and OpenAI Gym environments, do not show significant differences.

---

### Official Review · AnonReviewer2 · 2019-10-23
**Official Blind Review #2**

**Rating:** 1

**Review:**

The work contributes a library emulating Atari games in GPU in parallel and allowing to speed-up the execution of reinforcement learning algorithms.

I see that the paper qualifies to the conference; in particular there is listed the topic:

- “implementation issues, parallelization, software platforms, hardware”

However, this is not a research paper, and I do not really see how I should asses it. What I can say about it is that it is considerable amount of work, not only implementing the simulator but also looking at what RL methods need, and how to optimize the allocation and exchange of the data so that everything would work on GPU more efficiently.

From the practical perspective, I am somewhat confused. The speed-up factors in the experiments are rather modest: about 4x for simulating and rendering frames, 2.5x for full RL, on a single GPU. Better with scaling to multi-GPU systems. In Table 1 the total training time per resources used differs dramatically. However if I look at the lines with A2C it is about the same time with 100-200 CPU cores + 1 GPU versus 12 cores + 1 GPU. So this is about factor 10 in the resources, versus CPU parallelization probably suffering overheads.

It appears that the maximum steed-ups are achieved for a particular type of the reinforcement learning algorithms, and using it in a general case would give a modest improvement.

The paper itself consists of introduction, related work, 1 page overview of what it means to simulate the Atari games, and experiments. So it is mostly about measuring the speedups, with several implementations / platforms.

I tend to think that this work will not very much boost the research for new RL methods. It is limited to Atari games, mostly helps to sample-inefficient RL methods and if it helps, the speed-up factors are not of the order that would make experiments by the researchers otherwise impossible.

I would also give priority to theoretical contributions at ICLR. In the end, we all are using CUDA and cnDNN, but presentations about how they implement things are rather given at GPU computing conferences.




**Experience Assessment:**

I do not know much about this area.

**Review Assessment: Checking Correctness Of Derivations And Theory:**

N/A

**Review Assessment: Checking Correctness Of Experiments:**

I assessed the sensibility of the experiments.

**Review Assessment: Thoroughness In Paper Reading:**

I read the paper at least twice and used my best judgement in assessing the paper.

---

> ### Author Response · Authors · 2019-11-09
> **Answer to reviewer 2**
>
> The comments from R2 and R1, who had to read the paper multiple times to appreciate his content (“My first reaction to this paper was, "So what?"; but as I read more, I like the paper more and more”), suggest that we could explain our research contributions in a better way.  Therefore we modify the Introduction to better highlight that we introduce CuLE as a tool to “both investigate and mitigate” the limitations of the existing DRL approaches, including the bottleneck analysis (contribution (1), mentioned as significant by R1), the batching strategy (contribution (2), mentioned as significant by R3), as well as the analysis of the advantages and limitations of GPU-emulation (contribution (3), mentioned as significant by R1) that eventually lead to open research questions (see the Conclusion, mentioned as significant by R3). Our paper follows the track of other works on efficient RL with additional attention to system details (e.g. Babaeizadeh et al. (2016; 2017); Stooke & Abbeel (2018a); Espeholt et al. (2018)) and, more broadly speaking, of many works aimed at making ML computationally efficient, considering not only the algorithmic aspects, but also communication issues and system level optimizations (e.g., Seunghak Lee et al.,  On Model Parallelization and Scheduling Strategies for Distributed Machine Learning, NIPS 14; Peng Jiang et al., A Linear Speedup Analysis of Distributed Deep Learning with Sparse and Quantized Communication, NeurIPS 18; Michael Teng et al., Bayesian Distributed Stochastic Gradient Descent, NeurIPS 18; Jianqiao Wangni et al., Gradient Sparsification for Communication-Efficient Distributed Optimization, NeurIPS 18; ...).
>
> As for R2’s point on giving priority to theoretical contributions, we believe that a combination of algorithmic innovations and system research (as in Babaeizadeh et al. (2016; 2017); Stooke & Abbeel (2018a); Espeholt et al. (2018)) is necessary to advance the state of the art in DRL. Coherently with this, we mention in the first paragraph of Related Work the two factors affecting the wall clock convergence time of a DRL algorithm: sample and computational efficiency. R2 notices that “the maximum speed-ups are achieved for a particular type of reinforcement learning algorithms, and using it (CuLE) in a general case would give a modest improvement,” which is indeed highlighting once more the strong interaction between a DRL algorithm and its implementation, and the need to study these two aspects together. Partially coherently with R2’s claim that CuLE “in a general case would give a modest improvement”, we do show that vanilla A2C does not benefit from CuLE’s high data throughput (see Table 3 and Figure 7), but through an ad hoc batching strategy (Fig. 6) we achieve a 2-4x speed up in terms of convergence time. On the other hand, we believe that CuLE (and GPU emulation/simulation in general) can be beneficial for a wider class of algorithms. We demonstrate (Table 5) that DQN and PPO achieve a significant speed up (in terms of raw frames per second) using CuLE, and almost linear scaling on multiple GPUs (Table 4). Further speed up can be obtained by optimizing the GPU mapping of the TIA kernel (end of the second paragraph, Section 3), or by memory compression (second paragraph, Discussion and Conclusion). Thus we demonstrate that at least the computational efficiency aspect can be improved for different classes of algorithms. The problem of leveraging the high data throughput generated by the GPU using these (and potentially others) algorithms, to achieve high sample efficiency and fast convergence at the same time, remains open, and we believe that CuLE represents a valid tool for future investigations in this direction. To this aim, the Atari suite contains a set of tasks that are sufficiently hard to be non-trivial (as discussed in the Reinforcement Learning Workshop, ICML 2017), and, at the same time, sufficiently simple to be solved in a reasonable amount of time.
>
> The magnitude of the speed-up for A2C+V-trace is comparable with results reported in DRL literature - the fact that it is not as impressive as the ones achieved in supervised learning in the last few years, further demonstrates the high complexity of the problem of accelerating DRL, which cannot be regarded as a pure algorithmic problem, but requires the parallel investigation of both the algorithmic and system implementation details.
>
> To better justify the description of the CuLE’s implementation in Section 3, we add a paragraph at the end of the manuscript to highlight that “CuLE’s implementation is informed by ease of debugging, need for flexibility, and compatibility with standard DRL benchmarks. These choices put a limit on the achievable speed up factor (for instance by using emulation of the Atari 2600 console instead of direct CUDA implementations of Atari games), but the analysis and insights provided in our paper furnish indications for the design of efficient simulators for DRL”.

---

### Official Review · AnonReviewer1 · 2019-10-24
**Official Blind Review #1**

**Rating:** 8

**Review:**

This paper describes a port of the Atari Learning Environment to CUDA, reports on a set of performance comparison, and provides a bottleneck analysis based communication bandwidth and various throughputs required to saturate them for training and inference.

My first reaction to this paper was, "So what?"; but as I read more, I like the paper more and more.  It was the bottleneck analysis that changed my mind.  It was done very thoroughly and it provides deep insight in the challenges that RL faces for both learning and inference in a variety of settings.  I especially liked the analysis of the advantages and limitations of GPU emulation.  I also thought the Discussion section was well written.

The paper would be better if:
1) The figure fonts were larger throughout the paper.
2) The gaps in Table 1 were explained.

Minor issue:  Change "feed" to "fed" on page 3.


**Experience Assessment:**

I have read many papers in this area.

**Review Assessment: Checking Correctness Of Derivations And Theory:**

I assessed the sensibility of the derivations and theory.

**Review Assessment: Checking Correctness Of Experiments:**

I assessed the sensibility of the experiments.

**Review Assessment: Thoroughness In Paper Reading:**

I read the paper at least twice and used my best judgement in assessing the paper.

---

> ### Author Response · Authors · 2019-11-08
> **Answer to reviewer 1**
>
> We enlarged the figures and used larger fonts in the paper to improve readability. Data in the first two sections of Table I have been taken from Table I in Horgan et al., 2018, and completed with the frames per seconds generated and consumed by each different method, when available in the corresponding paper (see caption of Table I on our paper). For the last two sections of Table I, that are fully completed, we used data from Stooke and Abbeel, 2018b, Espeholt et al., 2018, and our own experiments with CuLE; we updated Table I to use N/A to indicate meaningless metrics (e.g. training time for CuLE in emulation only mode). The typo on page 3 has been fixed. We thank the reviewer for the comments.

---

### Decision · Program_Chairs · 2019-12-19

**Decision:**

Reject

**Comment:**

The paper presented a detailed discussion on the implementation of a library emulating Atari games on GPU for efficient reinforcement learning. The analysis is very thoroughly done. The major concern is whether this paper is a good fit to this conference. The developed library would be useful to researchers and the discussion is interesting with respect to system design and implementation, but the technical depth seems not sufficient.